## [Decision Letter · Decision Letter 0]

8 Dec 2025

Dear Dr. Chu,

Thank you for submitting your manuscript to PLOS ONE. After careful consideration, we feel that it has merit but does not fully meet PLOS ONE’s publication criteria as it currently stands. Therefore, we invite you to submit a revised version of the manuscript that addresses the points raised during the review process.

We look forward to receiving your revised manuscript.

Kind regards,

Dr. Ahmed El-Fiqi, Ph.D.

Academic Editor

PLOS One

Journal Requirements:

2. To comply with PLOS One submissions requirements, in your Methods section, please provide additional information regarding the experiments involving animals and ensure you have included details on (1) methods of sacrifice, and (2) efforts to alleviate suffering.

Reviewers' comments:

Reviewer's Responses to Questions

**Comments to the Author**

1. Is the manuscript technically sound, and do the data support the conclusions?

Reviewer #1: Partly

2. Has the statistical analysis been performed appropriately and rigorously?

Reviewer #1: Yes

3. Have the authors made all data underlying the findings in their manuscript fully available?

Reviewer #1: Yes

4. Is the manuscript presented in an intelligible fashion and written in standard English?

Reviewer #1: Yes

Reviewer #1: Summary

This study addresses an important question related to age-associated impairment of cartilage repair and combines single-cell RNA-seq with in vivo AAV–Arg-1 overexpression experiments to propose Arg-1 as a key regulator that improves cartilage repair in aged animals. The overall direction is of interest, the general experimental framework is coherent, and the technical workflow is, in principle, appropriate. However, to reach a publishable standard, the manuscript requires substantial strengthening in terms of data rigor, analytical robustness, mechanistic interpretation, figure quality, and the consistency between results and conclusions. Overall, this is a potentially valuable study, but in its current form it remains largely at the level of “phenotypic observations plus relatively basic functional validation.” The mechanistic depth regarding Arg-1 is insufficient, and some interpretations of the single-cell data appear overstated or not fully supported by the presented evidence.

Major Comments

1. The authors have performed basic clustering, DEG, and GO analyses, but several central conclusions—such as the statement that “aged animals fail to differentiate into anti-inflammatory macrophages”—are not sufficiently supported. I recommend: Performing formal cell proportion tests (e.g., χ² tests, logistic regression, or dedicated tools such as scCODA) to demonstrate that differences in macrophage subset frequencies are statistically significant across conditions. Providing an assessment of trajectory stability in the pseudotime analysis (e.g., testing alternative root cell definitions and/or bootstrapping to show robustness of the inferred branching structure). Supporting UMAP-based interpretations with quantitative metrics (e.g., subset frequencies, module scores, or distance measures), rather than relying primarily on qualitative descriptions of embeddings. These additional analyses would substantially increase the credibility of the single-cell component.

2. The manuscript repeatedly presents Arg-1 as a “central regulatory node” in macrophage polarization and cartilage repair. However, the current data essentially show: Arg-1 overexpression, reduced ROS levels, and increased M2 marker expression (by RT-qPCR). What is missing includes: Functional assays of macrophage behavior (e.g., phagocytosis, chemotaxis, cytokine/chemokine secretion profiles). A loss-of-function or inhibition approach (e.g., pharmacologic inhibition, knockdown, or knockout) to test whether Arg-1 is necessary for the observed protective effects (rescue or reverse experiments). Direct evidence that aged macrophages differ in their responsiveness to Arg-1 compared with young macrophages. I strongly recommend aligning the strength of the conclusions with the strength of the data. Without additional mechanistic experiments, Arg-1 should be described more cautiously as a promising modulator or contributor, rather than as a fully established central regulator.

3. Given PLOS ONE’s emphasis on rigor and reproducibility in animal studies, the description of the in vivo design needs to be substantially improved. Please clarify: How animals were randomly allocated to groups. Whether histological scoring (e.g., Mankin or modified Mankin scores) and other outcome assessments were performed blinded to treatment group. Whether sample size calculations (power analyses) were performed a priori, or, if not, provide a rationale for the chosen group sizes. Whether any animals were excluded, died, or replaced during the experiment, and how such events were handled in the analysis. Without these details, the robustness of the in vivo findings is difficult to evaluate.

4. The current histology results are described mostly in qualitative terms. For a convincing demonstration of structural protection or regeneration, quantitative analyses are needed. I recommend: Including OARSI or modified Mankin scores (or another standardized histological scoring system) for articular cartilage degeneration. Quantifying relevant structural parameters, such as cartilage thickness, subchondral bone changes, and/or proteoglycan content (e.g., Safranin O density), using standardized image analysis. Performing blinded image quantification (e.g., using ImageJ or QuPath) and reporting the scoring methods and inter-observer agreement if multiple observers are involved. These additions would markedly strengthen the histological evidence for Arg-1–mediated protection.

5. The manuscript interprets the Cdca-high cluster as “pro-inflammatory macrophages.” However: CDCA-related genes are more commonly associated with cell cycle and proliferative states, and are not canonical macrophage markers. This cluster may represent proliferating cells (including proliferative macrophages) or even doublets / multiplets, rather than a distinct pro-inflammatory macrophage subset. To clarify this, the authors should: Apply doublet-detection tools (e.g., DoubletFinder, Scrublet) to exclude the possibility that this cluster is driven by technical artifacts. Confirm that Cdca-high cells express macrophage hallmark genes (e.g., Lyz2, Csf1r, Adgre1), rather than predominantly cell cycle markers. Re-evaluate the biological interpretation of this cluster in light of these findings, and moderate the related conclusions if necessary. Without such analyses, the current interpretation of Cdca-high macrophages as a distinct pro-inflammatory subset appears speculative.

6. The manuscript repeatedly highlights the canonical competition between Arg-1 and iNOS for L-arginine and links this to inflammation resolution. While this is supported by the literature, the present study does not directly measure: L-arginine consumption, NO production or iNOS activity, or downstream NF-κB signaling inhibition. Given the lack of direct experimental evidence in this work, the authors should either: Add targeted experiments to support this mechanistic axis in their specific model, or Tone down the mechanistic statements and present the Arg-1–iNOS relationship as a plausible, literature-based hypothesis rather than a demonstrated mechanism in this study.

7. The Methods section provides extensive cloning and packaging details for the AAV8 vector but remains incomplete in terms of essential quality control parameters. I suggest: Providing data or at least a brief description of AAV vector purity (e.g., silver staining, dot blot, or capsid protein assessment). Detailing the titer determination method, including standard curve construction, Ct range, and how viral genomes per mL were calculated. Including a quantitative assessment of transduction efficiency beyond representative GFP images (e.g., percentage of GFP⁺ cells in target tissues by flow cytometry or standardized image quantification).This will help readers assess the robustness and reproducibility of the gene delivery platform.

8. The background section currently lacks sufficient citation support. To more effectively contextualize the study and highlight the rapidly growing importance of single-cell omics in immunology and tissue repair research, the authors should expand their introduction with key recent advances enabled by single-cell technologies. In particular, it is recommended to incorporate and discuss several influential studies that underscore: the field’s increasing reliance on single-cell omics, the new mechanistic insights these technologies have revealed, and their relevance to immune regulation and tissue remodeling. Suggested references include:

(1) https://doi.org/10.1002/imt2.132

(2) https://doi.org/10.1002/imt2.217

(3) https://doi.org/10.1002/imt2.40

(4) https://doi.org/10.1002/imt2.117

(5) https://doi.org/10.1002/imt2.226

For macrophage biology and macrophage-mediated mechanisms, the following reference is also recommended:

(6) https://doi.org/10.1002/imt2.233

Incorporating these studies will help readers better appreciate current developments in the field and strengthen the conceptual foundation of the manuscript.

Minor Comments

1.Figure legends are generally too brief and should more clearly describe experimental conditions, sample sizes, statistical tests, and key readouts—especially for Figures 2, 4, and 7.

2.Figure 1 lacks a description of single-cell data preprocessing and quality control procedures.

3.Immediately after Figure 1A, the manuscript should provide quantitative statistics of cell-type composition across the four experimental groups, as these comparisons are essential to interpret differences between control and injury/age conditions.

4.Figures 1C–D do not adequately demonstrate statistical significance, and the same issue applies to Figures 2B–C; appropriate statistical tests or visualization methods should be added.

5.The rationale for selecting the marker genes shown in Figure 2D is unclear.

6.In Figure 2E, the justification for choosing the pseudotime starting point is not provided. Furthermore: Homeostatic macrophages appear at both the beginning and end of the trajectory—how is this contradiction interpreted? The Intermediate state lies along both branching paths—does this indicate potential subtypes or transitional sub-states that require further characterization?

7.The basis for selecting the gene sets used in Figure 3A is not clearly explained.

8.In Figure 5, the fluorescence intensity of the Arg-1-UR + LPS group appears lower than the baseline control, which is biologically unexpected because Arg-1 overexpression should attenuate LPS-induced ROS but should not reduce ROS levels below physiological baseline. This raises the possibility of (i) inconsistent fluorescence exposure settings between groups or (ii) Arg-1-mediated suppression of probe oxidation rather than true ROS depletion. I recommend clarifying the imaging acquisition settings, confirming that all groups received identical LPS stimulation, and, ideally, validating ROS levels using a quantitative assay (e.g., flow cytometry or MitoSOX).

9.Figure 7 provides in vivo histological evidence of Arg-1–mediated protection after cartilage injury; however, several aspects require clarification to ensure consistency with earlier mechanistic data, especially the ROS findings in Figure 5. (1) Quantification Needed. The histology panels are fully qualitative. To support the interpretation, please include: OARSI or modified Mankin scores, Safranin O intensity quantification, Cartilage thickness / subchondral bone metrics, Blinded scoring. These measures are necessary to substantiate the visual observations. (2) Inconsistency with Figure 5 Expectations. Based on Figure 5, Arg-1-UR should partially rescue injury-induced degeneration, but should not surpass normal (Ctrl) morphology. Yet in Figure 7, Arg-1-UR appears more intensely stained and structurally “better” than Ctrl. This raises concerns about: inconsistent staining/exposure settings, non-comparable tissue regions, or technical artifacts. Please confirm standardized imaging conditions and ensure comparable anatomical sites across groups. (3) Apparent Contradiction Across Groups. Arg-1-UR showing stronger Safranin O staining and denser collagen than Ctrl is biologically unlikely. Potential explanations (staining batch effects, sampling bias, exposure variability) should be addressed, and controls provided to rule out technical artifacts. (4) Missing Mechanistic Link to Figure 5. The manuscript should explicitly connect the in vitro ROS suppression (Figure 5) to the in vivo ECM preservation shown in Figure 7, and clarify that Arg-1 is not expected to create cartilage exceeding normal baseline.

I would like to sincerely thank the editor for the opportunity to review this carefully executed and clinically relevant manuscript. The authors investigate an important yet understudied dimension of cartilage biology—how age-related immune dysregulation, particularly macrophage polarization dynamics, shapes tissue repair outcomes. The integration of single-cell transcriptomics with in vivo Arg-1 overexpression provides a conceptually appealing framework that has the potential to deepen our understanding of age-dependent regenerative decline and the immunological determinants of cartilage healing.

I hope that my comments will be helpful in sharpening the analytical rigor, refining the mechanistic claims, and strengthening the quantitative evidence underlying the main conclusions. I look forward to seeing a revised version, and I believe that with these substantive improvements, this study has the potential to make a meaningful contribution to the fields of osteoarthritis biology, macrophage immunology, and age-impaired tissue regeneration.

**Do you want your identity to be public for this peer review?** For information about this choice, including consent withdrawal, please see our Privacy Policy

Reviewer #1: No

---

## [Author Response · Author response to Decision Letter 1]

25 Jan 2026

We sincerely thank the reviewer for their thoughtful and constructive evaluation of our manuscript, as well as for recognizing the importance of the question and the overall interest of the study direction. We appreciate the positive comments on the experimental framework and technical workflow. We fully agree with the reviewer's assessment that the manuscript requires strengthening in several key areas to reach a publishable standard. We have taken all points to heart and have conducted substantial additional experiments, in-depth data re-analyses, and extensive revisions to the text and figures to address the concerns regarding data rigor, analytical robustness, mechanistic depth, figure quality, and the alignment of conclusions with evidence. A detailed, point-by-point response to each comment from the reviewers and the editor is provided in the separate file 'Response to Reviewers'. All changes made to the manuscript have been highlighted in the 'Revised Manuscript with Track Changes' file.

---

## [Decision Letter · Decision Letter 1]

1 Feb 2026

Dear Dr. Chu,

Thank you for submitting your manuscript to PLOS ONE. After careful consideration, we feel that it has merit but does not fully meet PLOS ONE’s publication criteria as it currently stands. Therefore, we invite you to submit a revised version of the manuscript that addresses the points raised during the review process.

We look forward to receiving your revised manuscript.

Kind regards,

Dr. Ahmed El-Fiqi, Ph.D.

Academic Editor

PLOS One

**Journal Requirements:**

Reviewers' comments:

Reviewer's Responses to Questions

**Comments to the Author**

Reviewer #1: (No Response)

2. Is the manuscript technically sound, and do the data support the conclusions?

Reviewer #1: Partly

3. Has the statistical analysis been performed appropriately and rigorously?

Reviewer #1: No

4. Have the authors made all data underlying the findings in their manuscript fully available?

Reviewer #1: Yes

5. Is the manuscript presented in an intelligible fashion and written in standard English?

Reviewer #1: No

Reviewer #1: The authors have addressed most of the questions and concerns, but several key issues remain unresolved and some details still need to be clarified.

1. The cell-number comparisons across the four groups in Figure 1B are not meaningful, because these differences are likely driven by sampling depth rather than biology. The authors may have misunderstood part of my previous comment. Similarly, comparing absolute cell numbers across groups in Figure 1B–C and Figure 2B–C is not statistically meaningful, because the total number of captured cells largely reflects how many cells were loaded/retained per sample (i.e., sampling variation). Put simply, if one group was sampled more deeply, it will naturally contain more cells.

2. The comparisons in Figure 1D and Figure 2D are meaningful, but the analysis is incomplete. The authors only present a subset of the differential-testing results. Please provide the full set of statistical comparisons (covering all relevant groups and subsets), and clearly describe the statistical methods in detail (test type, unit of analysis, multiple-testing correction, and any covariates or batch/donor structure if applicable).

3. If the authors want to emphasize group differences in Figure 2E, statistical testing must be added. Please specify what is being tested, which groups are compared, what test is used, and how multiple comparisons (if any) are controlled.

4. The statistical approach used for Figure 3D is unclear. Please explicitly state the statistical unit (cell vs. animal), the test/model used, how many biological replicates contribute to each group, and whether any adjustments (e.g., mixed-effects modeling or pseudo-bulk aggregation) were applied.

5. Figure 4A is not sufficiently clear about the in vivo implantation procedure. Please clarify the implantation method and the exact anatomical site/region, and revise the schematic so that the implantation route and location are visually explicit and immediately interpretable from the figure itself.

6. The writing and formatting of the manuscript should be strengthened. Some figure labels and the main text are inconsistent or do not follow common standards (e.g., “b cell” in Figure 1 is incorrect). In addition, descriptions of cell types alternate between “cell” and “cells” in a non-systematic way, which is not rigorous. Similar issues appear throughout and should be carefully standardized and corrected.

**Do you want your identity to be public for this peer review?** For information about this choice, including consent withdrawal, please see our Privacy Policy

Reviewer #1: No

---

## [Author Response · Author response to Decision Letter 2]

9 Feb 2026

Reviewer #1: The authors have addressed most of the questions and concerns, but several key issues remain unresolved and some details still need to be clarified.

We sincerely thank the reviewer for their constructive feedback and careful review of our revised manuscript. We are pleased that most of the previous concerns have been addressed, and we appreciate the opportunity to further clarify the remaining points. Below, we provide point-by-point responses to the outstanding issues, with corresponding revisions detailed for each comment. All suggested changes have been incorporated into the manuscript to ensure clarity, rigor, and completeness of the presented work.

1. The cell-number comparisons across the four groups in Figure 1B are not meaningful, because these differences are likely driven by sampling depth rather than biology. The authors may have misunderstood part of my previous comment. Similarly, comparing absolute cell numbers across groups in Figure 1B–C and Figure 2B–C is not statistically meaningful, because the total number of captured cells largely reflects how many cells were loaded/retained per sample (i.e., sampling variation). Put simply, if one group was sampled more deeply, it will naturally contain more cells.

We agree with the reviewer’s point. Absolute cell numbers can indeed be influenced by technical variation in sampling depth rather than genuine biological differences. Therefore, as suggested, we have removed all figures and analyses comparing absolute cell numbers across groups. Instead, we have retained and focused on cell proportion comparisons (relative frequencies), which are more appropriate for assessing biologically relevant changes.

2. The comparisons in Figure 1D and Figure 2D are meaningful, but the analysis is incomplete. The authors only present a subset of the differential-testing results. Please provide the full set of statistical comparisons (covering all relevant groups and subsets), and clearly describe the statistical methods in detail (test type, unit of analysis, multiple-testing correction, and any covariates or batch/donor structure if applicable).

Thank you for this suggestion. We have now performed and included full pairwise statistical comparisons for all relevant groups and cell subsets in the revised figures (Figure 1D and 2D). The analysis was conducted on cell proportions derived from three biological replicates per group. We used Tukey’s Honestly Significant Difference (HSD) test following one-way ANOVA to control for multiple comparisons. No batch or donor covariates were included, as the experiment did not involve batch effects or paired donor structure. A detailed description of the statistical methods has been added to the Methods section. Line 165-179, line 407-414, line 446-455.

3. If the authors want to emphasize group differences in Figure 2E, statistical testing must be added. Please specify what is being tested, which groups are compared, what test is used, and how multiple comparisons (if any) are controlled.

We have revised Figure 2E to retain only the cell subsets most relevant to the study conclusions. Because the gene expression data for certain cell groups were limited and did not meet normality assumptions, we did not use ANOVA. Instead, we performed Kruskal–Wallis tests for each cell subset to compare expression levels across groups, followed by Dunn’s post‑hoc test with Benjamini–Hochberg correction for multiple comparisons. The specific groups compared, test used, and correction method are now clearly stated in the figure legend and Methods. Line 154-164, line 455-458

4. The statistical approach used for Figure 3D is unclear. Please explicitly state the statistical unit (cell vs. animal), the test/model used, how many biological replicates contribute to each group, and whether any adjustments (e.g., mixed-effects modeling or pseudo-bulk aggregation) were applied.

We apologize for the lack of clarity. Figure 3D presents semi‑quantitative Western blot data. The statistical unit is the animal (n = 3 biological replicates per group). Data were analyzed using Student’s t-test. No mixed‑effects or pseudo‑bulk adjustments were applied because the experimental design did not involve repeated measures or nested sampling. These details have been explicitly added to the Methods section. Line 234-244.

5. Figure 4A is not sufficiently clear about the in vivo implantation procedure. Please clarify the implantation method and the exact anatomical site/region, and revise the schematic so that the implantation route and location are visually explicit and immediately interpretable from the figure itself.

Thank you for pointing this out. Figure 4A primarily illustrates the in vitro culture process. The in vivo implantation and evaluation procedures are detailed separately in Figure 8. Following the reviewer’s suggestion, we have revised both schematic diagrams to more clearly depict the implantation route, anatomical site, and experimental timeline, making them visually explicit and easier to interpret.

6. The writing and formatting of the manuscript should be strengthened. Some figure labels and the main text are inconsistent or do not follow common standards (e.g., “b cell” in Figure 1 is incorrect). In addition, descriptions of cell types alternate between “cell” and “cells” in a non-systematic way, which is not rigorous. Similar issues appear throughout and should be carefully standardized and corrected.

We sincerely apologize for these inconsistencies. We have now systematically reviewed and corrected all figure labels, nomenclature, and terminology throughout the manuscript. Specifically: “b cell” has been corrected to “B cells” (capitalized, as per standard immunology nomenclature). Similar formatting and labeling issues in other figures and the text have been standardized to ensure rigor and consistency.

We believe these revisions have significantly improved the clarity and professionalism of the manuscript. Thank you again for your valuable comments.

---

## [Decision Letter · Decision Letter 2]

25 Feb 2026

Single-Cell Omics Reveals Arg-1 as a Key Regulator of Age-Dependent Macrophage-Mediated Cartilage Repair

PONE-D-25-55813R2

Dear Dr. Chu,

We’re pleased to inform you that your manuscript has been judged scientifically suitable for publication and will be formally accepted for publication once it meets all outstanding technical requirements.

Kind regards,

Dr. Ahmed El-Fiqi, Ph.D.

Academic Editor

PLOS One

Additional Editor Comments (optional):

Please address the minor revisions recommended by  reviewer 1 during the correct proof  as follows:

The authors addressed or responded to all concerns and comments. However, the clarity of the illustrations (such as images in the cell experiment section and other related results, Figures 6\7\8) should be further improved. Furthermore, all raw data should be publicly available or accessible (including, but not limited to, sequencing data, raw data from wet experiments, etc.).

Reviewers' comments:

Reviewer's Responses to Questions

**Comments to the Author**

Reviewer #1: All comments have been addressed

2. Is the manuscript technically sound, and do the data support the conclusions?

Reviewer #1: Yes

3. Has the statistical analysis been performed appropriately and rigorously?

Reviewer #1: Yes

4. Have the authors made all data underlying the findings in their manuscript fully available?

Reviewer #1: Yes

5. Is the manuscript presented in an intelligible fashion and written in standard English?

Reviewer #1: Yes

Reviewer #1: The authors addressed or responded to all concerns and comments. However, the clarity of the illustrations (such as images in the cell experiment section and other related results, Figures 6\7\8) should be further improved. Furthermore, all raw data should be publicly available or accessible (including, but not limited to, sequencing data, raw data from wet experiments, etc.).

**Do you want your identity to be public for this peer review?** For information about this choice, including consent withdrawal, please see our Privacy Policy

Reviewer #1: No

---

## [Editor Report · Acceptance letter]

PONE-D-25-55813R2

PLOS One

Dear Dr. Chu,

I'm pleased to inform you that your manuscript has been deemed suitable for publication in PLOS One. Congratulations! Your manuscript is now being handed over to our production team.

Kind regards,

on behalf of

Dr. Ahmed El-Fiqi

Academic Editor

PLOS One